# Environmental and Economic Prioritization of Building Energy Refurbishment Strategies with Life-Cycle Approach

**Xabat Oregi** [1,*] , **Rufino Javier Hernández** [1] and **Patxi Hernandez** [2]

1 CAVIAR Research Group, Department of Architecture, University of the Basque Country UPV/EHU, 20018 San Sebastián, Spain; rufinojavier.hernandez@ehu.eus

2 Tecnalia, Basque Research and Technology Alliance (BRTA), 48160 Derio, Spain; patxi.hernandez@tecnalia.com

* Correspondence: xabat.oregi@ehu.eus; Tel.: +34-943-01-5891

**Abstract:** An increasing number of studies apply life-cycle assessment methodology to assess the impact of a new building or to prioritize between different building refurbishment strategies. Among the different hypotheses to consider during the application of this methodology, the selection of the impact indicator is critical, as this choice will completely change the interpretation of the results. This article proposes applying four indicators that allow analysing the results of a refurbishment project of a residential building with the life-cycle approach: non-renewable primary energy use reduction (NRPER), net energy ratio (NER), internal rate of return (IRR), and life-cycle payback (LC-PB). The combination of environmental and economic indicators when evaluating the results has allowed to prioritize among the different strategies defined for this case study. Furthermore, an extensive sensitivity assessment reflects the high uncertainty of some of the parameters and their high influence on the final results. To this end, new hypotheses related to the following parameters have been considered: reference service life of the building, estimated service life of material, operational energy use, conversion factor, energy price, and inflation rate. The results show that the NRPE use reduction value could vary up to −44%. The variation of the other indicators is also very relevant, reaching variation rates such as 100% in the NER, 450% in the IRR, and 300% in the LC-PB. Finally, the results allow to define the type of input or hypothesis that influences each indicator the most, which is relevant when calibrating the prioritization process for the refurbishment strategy.

**Keywords:** building energy refurbishment; environmental and economic life-cycle assessment; impact indicators; sensitivity analysis

## 1. Introduction

Numerous studies and reports show that life-cycle assessment (LCA) methodology is currently the best framework available to assess the potential environmental impacts of any activity [1–5]. Proofs of this are numerous review studies that focus on analysing, grouping, and comparing other previous works focused on evaluating the environmental and/or economic impact of the construction industry under the perspective of life cycle [6–18].

As can be seen in all these studies, the application of LCA methodology implies making decisions from the first stage of the process that will directly influence the evaluation process and the interpretation of results. Each study must decide on the type of functional unit ($m^2$, $m^2$/year, inhabitant, GJ/capita, and so on), life span or reference service life of the building (years), life-cycle inventory technique (process-based, hybrid, or input–output), life-cycle stages considered by the study, inventory database

used to get the information (U.S. Life Cycle Inventory, Ecoinvent, Gabi, and so on), or the consideration of uncertainties.

It should be noted that a large part of the aforementioned works focused on analysing LCA studies applied to new buildings. However, another critical area of the construction sector is energy refurbishment, which is becoming increasingly relevant mainly in countries where the level of new construction compared with the number of buildings with refurbished energy is very low.

When applying LCA methodology in buildings to be refurbished, along with all the aforementioned parameters, it is also necessary to consider two other aspects: definition of the scope of the study (environmental and/or economic) and the choice of indicators to interpret the results and select the refurbishment strategy. As can be seen in the summary in Table 1, the overview of the literature on indicators of refurbishment of buildings shows that most studies only consider environmental indicators, where mainly primary energy or global warming potential (GWP) are applied. In relation to the economic evaluation, although there are currently numerous studies that integrate the life-cycle cost (LCC) assessment when evaluating the impact of renewable technologies or energy systems [19–21], there are few studies that jointly apply the life-cycle approach to quantify and evaluate the environmental and economic impact of an energy refurbishment of a building.

**Table 1.** State-of-the-art of indicators applied in studies of energy refurbishment of buildings with the life-cycle approach.

| Study | Environmental Indicator | Economic Indicator |
|:-----:|:-----------------------:|:------------------:|
| [22] | Global Warming Potential (GWP), Acidification Potential (AP), Eutrification Potential (EP), Photochemical Ozone Creation Potential (POCP) | Payback time |
| [23] | Primary Energy (PE), Net Energy Ratio (NER) | |
| [24] | | Payback period (years) Ringgit |
| [25] | Embodied Energy (EE), GWP | € |
| [26] | Gross Energy Requirement (GER), GWP | |
| [27] | PE, GWP, AP, EP, POCP | |
| [28] | PE | |
| [29] | PE | |
| [30] | Cumulative Energy Demand (CED), GWP | |
| [31] | GWP | € |
| [32] | PE | € |
| [33] | GWP, PE | |
| [34] | GWP, PE | |
| [35] | GWP | |
| [36] | GWP, CED | |
| [37] | PE Reduction | Global cost reduction, investment cost reduction, simple payback period, discounted payback period, net present value |
| [38] | GWP | Present value, present cost |
| [39] | GWP, CED, AP, Ozone Depletion Potential (ODP), POCP, EP | |
| [40] | CED | Life-cycle cost |
| [41] | Operational Energy, GWP, AP, ODP, EP, Smog Formation | |

The objective of this article focuses on showing how the combination of four indicators allows the optimization of the prioritization process between a building's refurbishment strategies. For this, this work will propose the use of four indicators that will allow evaluating the environmental and economic influence of different building refurbishment strategies under the life-cycle perspective and optimizing the prioritization process of the strategies. On the one hand, the environmental behaviour of each strategy will be defined through the non-renewable primary energy use reduction (NRPER)

and net energy ratio (NER) indicators. On the other hand, the economic performance will be evaluated using the internal rate of return (IRR) and life-cycle payback (LC-PB) indicators.

An existing residential apartment building located in Donostia (Spain) is used to perform a detailed analysis. Furthermore, an exhaustive sensitivity analysis for a variety of parameters affecting the life-cycle assessment is also carried to discuss the influence of this uncertainty on the final results.

The paper is structured as follows. First, the methodology implemented in this study is described in Section 2. A case study used to test the proposed solution is described in Section 3. The results are shown and discussed in Section 4. A sensitivity and uncertainty evaluation is described in Section 5. Finally, the paper ends with conclusions.

## 2. Methodology

This paper follows a quantitative methodology to assess the impact generated at each stage in the life cycle of a building energy refurbishment project, following the EN15978:2011 [42] and EN16627:2015 [43] standards for environmental and economic performance evaluation.

### 2.1. Methodology

The standard EN 15978:2011 defines the scope of the different stages of the life cycle of a building: product stage (A1-3); construction process stage (A4-5); use stage (B1-7); end of life stage (C1-4); and benefits and load beyond the system boundary (D).

Despite the effort made to standardize the scope of each life-cycle stage, because of the lack of information, the complexity of the building, or the low impact of some stages compared with the total impact [44–46], very few studies have considered all the life-cycle stages [47–53]. All the studies analysed by this review considered the impact related to the product stage (A1-3) and operational energy use stage (B6). However, the number of studies that evaluated the other stages was lower: transport (A4) 83%, construction process (A5) 66%, end of life stage (C1-4) 62%, replacement (B4) 61%, and maintenance (B2) 46%. In relation to the stages that should be considered in a refurbishment project, the study of Oregi et al. [32] showed relatively minor importance of the transport (A4), construction process (A5), and end of life stages (C1-4). However, the relevance of the other life-cycle stages varied significantly depending on aspects such as the location, using the reference service life of the building.

In order to quantify the reduction of the impact (environmental and economic) of a refurbished building, the work methodology proposed by this study will be classified into five phases (see Figure 1). The first phase focuses on quantifying the impact of the non-refurbished building, which only considers the operation energy use stages impact of the current building. During the second phase, the impact associated with the product stage ($R_{A1-3}$) and construction process stage ($R_{A4-5}$) of each refurbishment strategy is quantified. The third phase evaluates the reduction of impact after applying each of these strategies ($B_{B6}$-$R_{B6}$). For this, it will be necessary to consider the impact linked to the maintenance stage ($R_{B2}$). When the estimated service life (ESL) value of the applied strategy is less than the reference service life (RSL) value of the refurbished building, the fourth phase quantifies the impact of the replacement stage ($R_{B4}$). Once the strategies are replaced, the renovated building will continue reducing its impact until reaching the limit of the RSL. At this end point, during the fifth phase, the impact of the end of life stage is quantified ($R_{C1-4}$). The evaluation of these five phases allows the obtained results to be grouped into two general groups. On the one hand, the reduction of the impact (environmental and economic) during the operational energy use stage with respect to the baseline scenario is quantified after applying the described strategies (positive impact, "Y"). On the other hand, the impact generated during the rest of the life-cycle stages is quantified after applying the strategies (positive impact, "X").

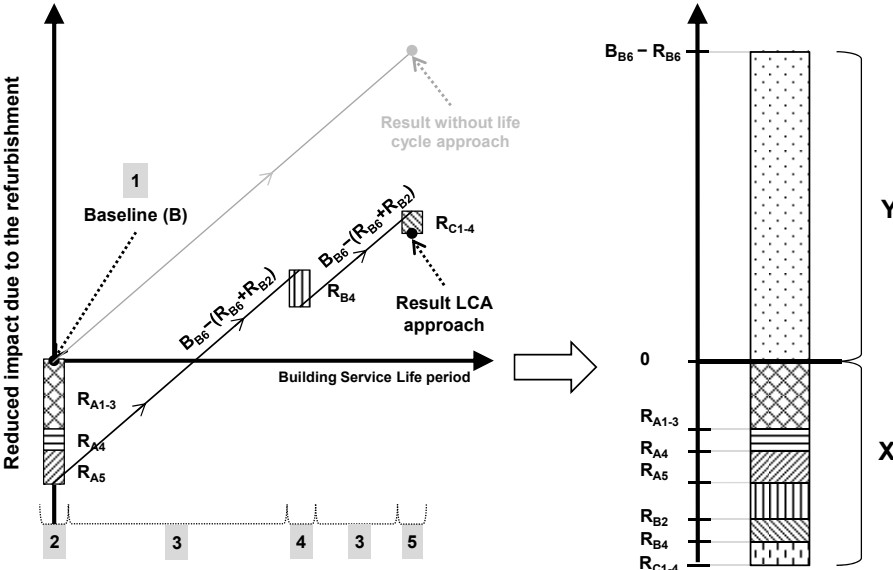

**Figure 1.** Diagram of the environmental and economic impact calculation of a refurbished building during its life cycle. LCA, life-cycle assessment.

### 2.2. Indicator

This work proposes the use of four indicators to prioritize between refurbishment strategies.

As can be seen in Table 1, many studies promote the use of other indicators such as global warming potential (GWP), which addresses the effect of increased temperature in the lower atmosphere. However, in order to evaluate only the impact linked to the use of non-renewable resources; this article decides to work with the non-renewable primary energy use reduction (NRPER) indicator. This indicator shows the difference on non-renewable primary energy use impact between the refurbished building (considering the impact related to all life-cycle stages) and the baseline building (considering only the current operational energy use impact).

The second environmental indicator selected by this study is the net energy ratio (NER). This indicator has been mainly used to evaluate technologies or processes related to the energy sector [54–58]. However, despite its potential, its use in the evaluation of refurbishment projects is very limited. The study of Hernández [23] was one of the first studies that applied this indicator to determine the energy efficiency of each refurbishment strategy during the life cycle of a refurbished building.

The assessment of this indicator is based on Equation (1). On the one hand, the reduction is calculated taking the difference of non-renewable primary energy use between the baseline building's annual operational energy use impact and the refurbished building's annual operational energy use impact ("Y" value of the Figure 1). On the other hand, considering the impact related to the product, construction process, replacement, and end of life stages, the annualized non-renewable primary energy use of each refurbishment strategy is calculated ("X" value of the Figure 1).

$$NER = \frac{\textit{Reduction of the annual non−renewable primary energy use during the operational stage}}{\textit{Annualized non − renewable primary energy use of each refurbishment strategy}} \quad (1)$$

Regarding the economic indicators, Table 1 shows some considered by previous studies. Additionally, there are other economic indicators that could also be considered in refurbishment project assessments: global cost [37], discount payback [37], net present value [37,59,60], net present cost [61], or the annual equivalent value cost [62]. After analysing the strength and the weakness of each one of them, this study proposes using two indicators to show the economic results of the different refurbishment scenarios: internal rate of return (IRR) and life-cycle payback (LC-PB), which allow evaluating the profitability and return period of each strategy.

Internal rate of return (IRR) is the interest rate at which the net present value (NPV) of all the cash flows (both positive and negative) from a project or an investment equals zero. The IRR is used to evaluate the attractiveness of a project or an investment. If the IRR of a new project exceeds a company's required rate of return, then the project is desirable. If IRR falls below the required rate of return, the project should be rejected. To calculate the IRR value of each strategy, this study has considered three types of economic impacts:

- Total investment (€/functional unit): This value considers economic impacts of three stages or phases. On the one hand, it considers the initial economic impact generated by each strategy during its product ($R_{A1-3}$) and construction process ($R_{A4-5}$) stages. On the other hand, this total investment also considers the impact of the replacement stage ($R_{B4}$). In order to calculate the impact of this stage, it will be necessary to multiply the current cost of each strategy by the value of the inflation rate for the year when it will be replaced. Finally, the total investment also considers the impact of the end of life stage ($R_{C1-4}$). As in the replacement stage, for the calculation of this value, it will be necessary to multiply the current cost associated with an end-of-life process by the inflation rate value of the year when the building reaches its life-period.
- Annual economic savings (€/functional unit): This value show the difference between the economic impact of the operational energy use stage before and after the refurbishment ($B_{B6}-R_{B6}$). Variations on the energy price directly influence this value.
- Maintenance cost (€/functional unit): This value considers the annual cost related to the maintenance of each refurbishment strategy ($R_{B2}$). It will be necessary to consider the inflation rate when calculating the impact of maintenance stage within the life-period of the refurbished building.

Many end users or residents find it difficult to interpret the percentage value offered by the IRR and link it to savings or profitability. As a result, in an economic analysis, this study proposes using the IRR indicator in conjunction with the life-cycle payback (LC-PB) indicator.

The payback period is the time it takes to cover investment costs. It can be calculated from the number of years elapsed between the initial investment ($R_{A1-3}$ and $R_{A4-5}$ stages), its subsequent operating costs, and the time at which cumulative savings offset the investment. Simple payback takes real (nondiscounted) values for future monies. Discounted payback uses present values. Payback, in general, ignores all costs and savings that occur after payback has been reached. When considering an investment with future expenditure, a discounted payback can be used to reflect the time value of money. It is possible that an investment with a short payback is a poorer option than one with a longer payback when looked at over the entire period of study. Owing to its simplified interpretation and ease of access for the end user, it is currently the most widely used economic indicator when it comes to prioritizing refurbishment strategies. However, as Figure 2 shows, when evaluating a refurbishment project from a life-cycle perspective, the calculation of the payback becomes more complex, generating scenarios with different interpretations.

The first scenario shows that, because of the initial investment and the impact of the replacement stage, the economic reduction of the building will be less than the impact generated. Therefore, there is no payback during the reference service life (RSL) of the building (see Figure 2a). The second scenario shows how, because of the reduced economic impact during the operational energy use stage and the low impact related to the replacement and end of life stages, there is one payback value during the RSL of the building (see Figure 2b). However, when the impact related to the replacement stage increases, is possible to obtain two different payback values: one related to the initial phase of the building and a second one after accounting for the replacement stage impact (see Figure 2c). Finally, it is also possible to obtain a scenario where, even though two different payback values are obtained during the reference service life of the building (before and after the replacement stages), the final result shows a negative value because of the impact of the end of life stage (see Figure 2d). That is, the impact generated is greater than the reduced impact.

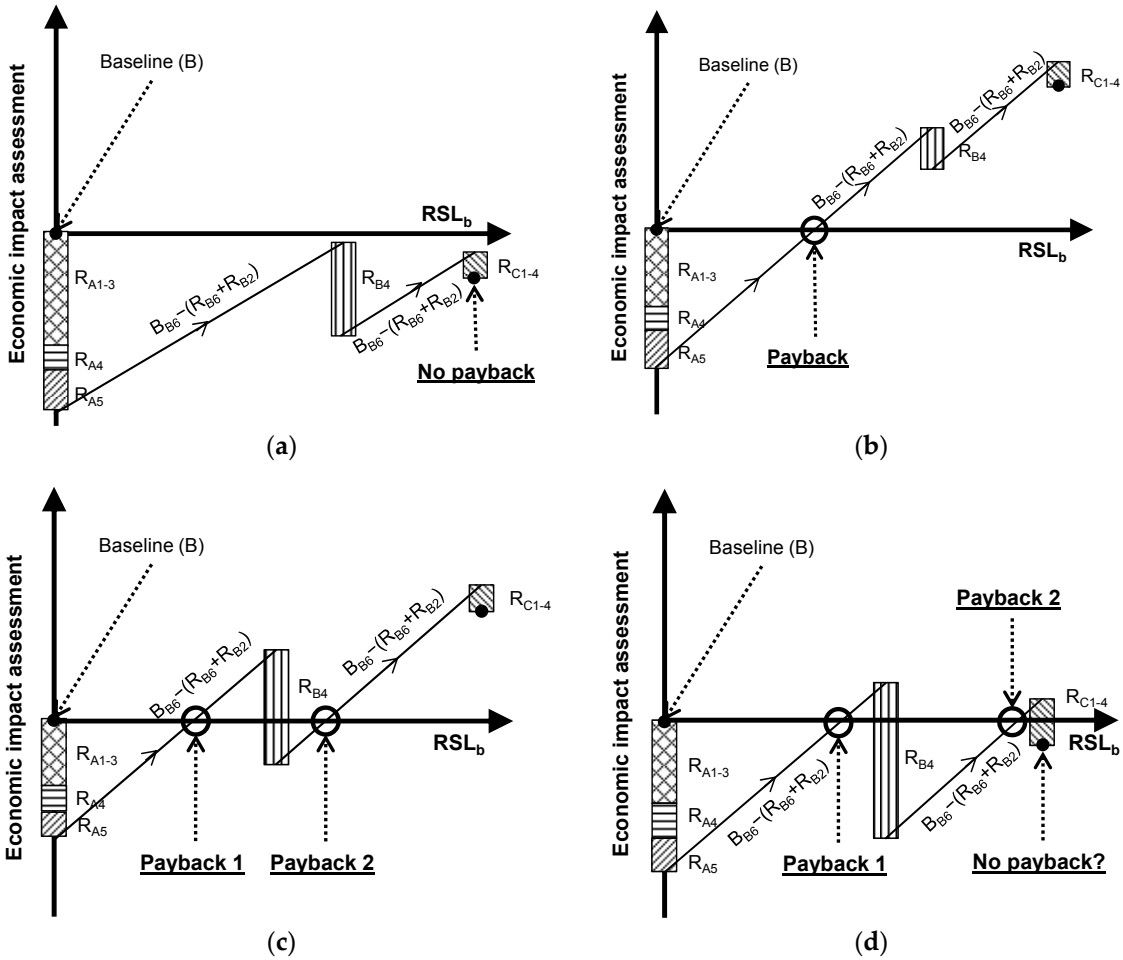

**Figure 2.** Scenarios when calculating the value of payback with a life-cycle approach: (**a**) no payback during the reference service life (RSL) of the building; (**b**) one payback value during the RSL of the building; (**c**) two different payback values; and (**d**) two paybacks during the RSL of the building, but with a final negative value.

This difficulty in applying the payback indicator when the study to be carried out considers the entire life cycle of the building to be refurbished, generating the need to adapt this indicator. Therefore, this study proposes to work based on the life-cycle payback (LC-PB) indicator, which allows evaluating the period of time (years) required to recover the total economic investment carried out during the different life-cycle stages of the refurbished building.

This new indicator is based on the concept of Figure 3. This new criterion proposes to group at the beginning of the evaluation the different negative impacts and economic investments that will be made during the life cycle of the refurbished building. That is, the total investment value considered also for the calculation of the IRR value. Once the starting point is defined, the annual impact reduction is then calculated. For this, annual economic savings ($B_{B6}-R_{B6}$) and maintenance cost ($R_{B2}$) will be considered (directly related to the variation of the energy price and the inflation rate values). In this way, when the savings are equal to the total investment, the end user will have the possibility of obtaining the life-cycle payback value of their refurbishment strategy.

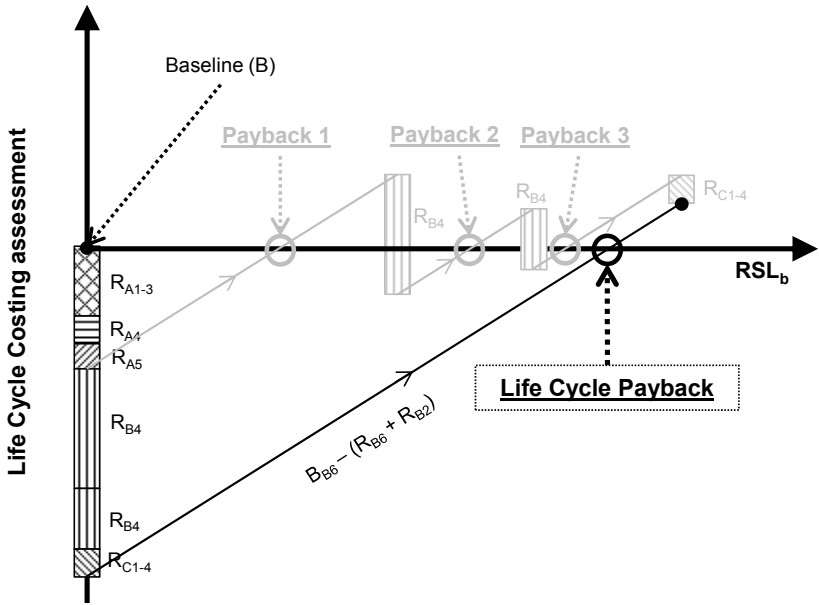

**Figure 3.** Scheme of the life-cycle payback (LC-PB) indicator calculation.

## 3. Case Study and Hypothesis

In order for the results obtained in this study to be replicated in existing buildings to a great extent, the scope of this study will focus on evaluating a particular type of building with high potential for energy refurbishment: residential buildings (multi-family) built between the years of 1960 and 1980. The selection of this type of building focuses on two aspects or factors:

- High percentage of residential buildings in the existing construction sector. The residential stock is the biggest segment with an EU floor space of 75% of the building stock [63].
- High percentage of the construction sector erected between the 1960s and 1980s. Within the existing European stock, a large share (more than 50%) is built before 1970s: Germany (73%), France (50%), United Kingdom (59%), Italy (69%), and Spain (46%) [64]. That is, before there were any or few requirements for energy efficiency and only a small part of these have undergone major energy retrofits, meaning that these have low insulation levels and their systems are old and inefficient.

In this case, the building selected for this study is the residential block located on Donostia (Spain). Table 2 summarizes the main aspects, hypothesis, and assumptions related to this case study.

**Table 2.** Summary of the main parameters related to the case study.

| Parameter | Case Study |
|---|---|
| Type of building | Residential building (multifamily) constructed in 1963 |
| Location, Climate zone | Donostia (Spain), Oceanic climate (Cfb) [65] |
| Geometric characteristics | Total net floor area of 9484 m$^2$; useful floor area of 8574 m$^2$<br>Ground floor + 9 residential floors |
| Current envelope thermal properties | Façade: 1.12 W/(m$^2$K); Deck: 2.34 W/(m$^2$K); First floor slab: 1.79 W/(m$^2$K); Monolithic glazing: 5.77 W/(m$^2$K); Aluminium frame: 4.2 W/(m$^2$K). |
| Heating system | Centralized natural gas boiler |
| Heating system setpoint | 21 °C |
| Heating, lighting, appliances, and occupancy schedules and values | Defined by the Spanish building regulations [66] |

**Table 2.** *Cont.*

| Parameter | Case Study |
|---|---|
| Domestic hot water system | Individual electric systems |
| Domestic hot water demand | 50 L/(person·day) |
| Ventilation (natural) | 0.75 r/h |
| Lighting: luminance level | 300 lux |
| Operation energy use assessment | Calculated by Design Builder software [67] |
| Proposed refurbishment strategies (a detailed description of each refurbishment strategy is available in Table A1 and Figure A1) | Window replacement (1); ventilated façade (2); external insulation system (3); internal insulation (4); air chamber insulation (5); solar thermal panels (6); photovoltaic panels (7); biomass boiler (8) |
| Considered life-cycle stages | Product stage ($R_{A1-3}$), Construction process stage ($R_{A4-5}$), Maintenance ($R_{B2}$), Replacement ($R_{B4}$), operational energy use ($B_{B6}$ and $R_{B6}$), and end of life stage ($R_{C1-4}$) |
| Functional unit | sq metre useful floor area |
| Reference service life of the building (RSL) | 50 years from the date of refurbishment |
| Estimated service life of refurbishment strategies (ESL) | Detailed information about the ESL value of each strategy is available in Table A2) |
| Indicators | Non-renewable primary energy use reduction (NRPER); net energy ratio (NER); internal rate of return (IRR); life-cycle payback (LC-PB) |
| Characterization method for environmental assessment | CML method [68] |
| Life-cycle inventory technique | Process-based |
| Inventory database and environmental impact of refurbishment strategies | Detailed information about the inventory databases and the environmental impact of each strategy is available in Table A2 |
| Economic impact of refurbishment strategies | Detailed information about the economic impact of each strategy is available in Table A3 |
| Transport process | 0.84 MJ/t·km (Articulated lorry (40 t) incl. fuel) [69] 0.00013 €/(km·kg) [70] |
| End of life stage. Landfill process | 0.18 MJ/kg (Landfill for inert matter, construction waste) [71] 0.07 €/kg [72] |
| End of life stage. Disposal, hazardous waste | 2.84 MJ/kg (Disposal, hazardous waste) [71] 0.12 €/kg [72] |
| Conversion factor | Natural gas: 1.23 MJ/MJ (Heat production, natural gas, at boiler modulating) [71]; Electricity: 1.74 MJ/MJ [73] |
| Energy price | Natural gas: 0.02 €/MJ; Electricity: 0.06 €/MJ [74] |
| Operational stage energy use price increment a year | Natural gas: 3%; Electricity: 4% [75] |
| Inflation rate | 1.5% [76] |

On the basis of the hypothesis and assumptions defined in Table 2, firstly, the environmental and economic impact of the baseline as well as the impact linked to each life-cycle stage after being refurbished by each strategy are calculated. As a second step, these impacts are transformed into the previously defined environmental (NRPER and NER) and economic (IRR and LC-PB) indicators, which will allow optimizing the prioritization process of the strategies.

## 4. Results and Discussion

There are different methods to visualize and interpret the results obtained (see Figure 4). The first option is separate visualization, where the environmental and economic impacts are shown separately [22,32]. Although the results are clearer to read, this option makes decision-making from a global perspective difficult. The second option is based on a weighting system, where a percentage value is applied to each indicator (environmental and economic), obtaining a final score that reflects the overall behaviour of each refurbishment strategy [27,77]. In this case, the different weighting criteria can

cause problems of subjectivity when prioritizing refurbishment strategies, as the weighting system is directly linked to a subjective process. Finally, the third option is the unified system [31,38]. This system allows the environmental and economic impacts to be shown in a single study (but separately), while avoiding the use of weighting systems.

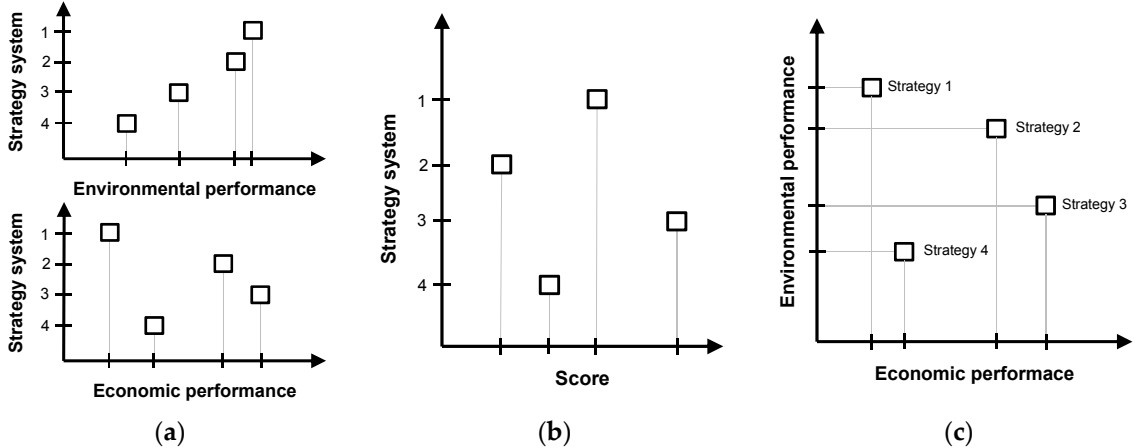

**Figure 4.** Scheme of the different methods to interpret the results: (**a**) separate method; (**b**) weighting method; (**c**) unified method.

On the basis of the unified method, Figures 5–8 reflect the results that shall be applied when prioritising and making decisions between different energy refurbishment strategies.

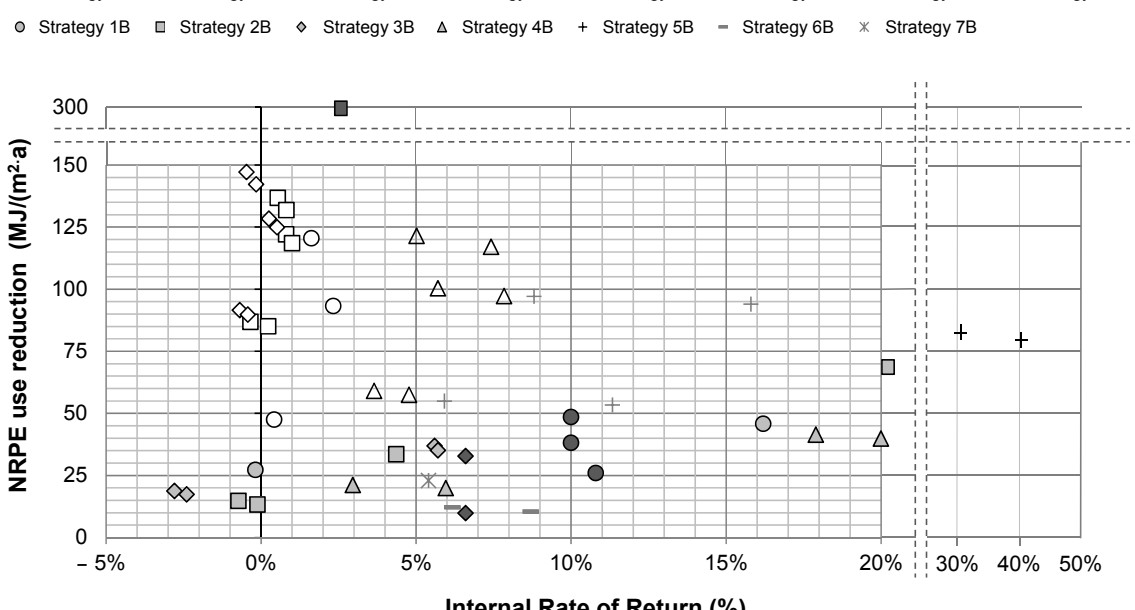

**Figure 5.** Analysis of the non-renewable primary energy use reduction (NRPER) and internal rate of return (IRR) values of each refurbishment strategy. Numeric values for this figure are available in Table A4.

The first overall reading of the results (see Figure 5) shows how, for those refurbishment strategies with a higher reduction rate of NRPE use, their IRR value is very low (even negative in many cases) and their LC-PB value is very high (see Figure 6). In fact, in some cases, even higher than the RSL value of the building to be refurbished (for this case study, a value of 50 years was considered). Ascione et al. [37] showed similar discounted payback period values, reflecting the relevance of the

life-cycle approach in this kind of assessments. However, other studies showed much lower payback values [78,79]. After analysing the hypothesis considered by these studies, it is relevant to define that both studies considered only the operational energy stage's impact reduction, without considering the impact linked to the other life-cycle stages.

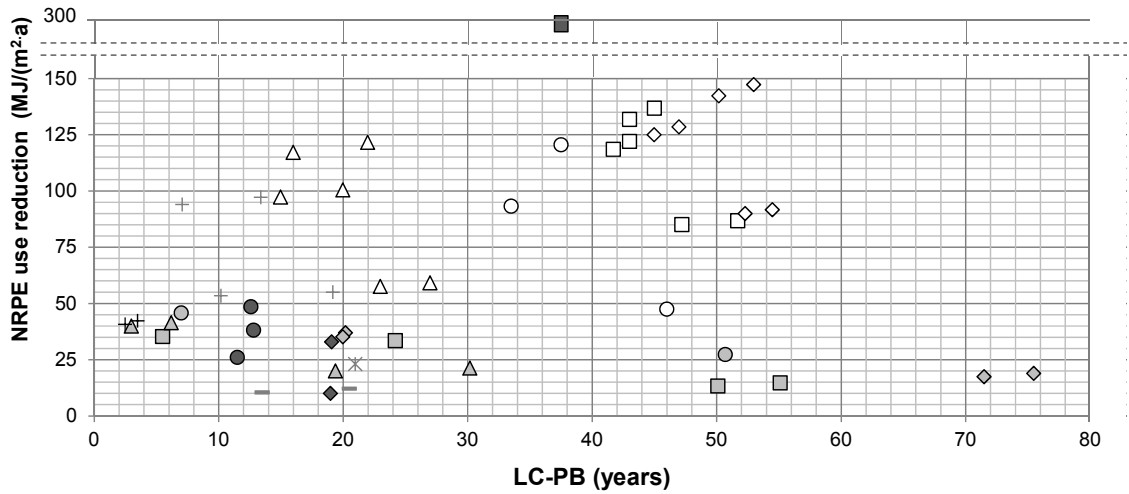

**Figure 6.** Analysis of the NRPER and life-cycle payback (LC-PB) values of each refurbishment strategy. Numeric values for this figure are available in Table A4.

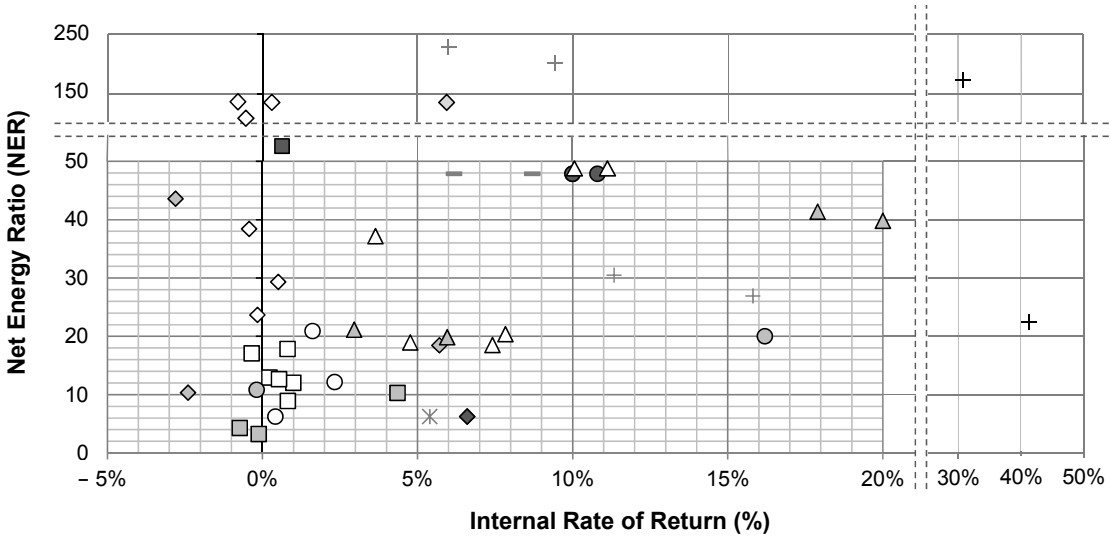

**Figure 7.** Analysis of the net energy ratio (NER) and IRR values of each refurbishment strategy. Numeric values for this figure are available in Table A4.

Even if refurbishment strategies such as the installation of ventilated facades or an external insulation system ("2" and "3") did reduce the existing building NRPE use overall impact by 20% during its life cycle, in most cases, their profitability is negative, reaching a value of 1% at best. That is, in a similar scenario to the one suggested in this case study, economic profitability for this type of refurbishment strategies is less than 3% of the advised profitability. This low profitability is directly

linked to the LC-PB value, as shown by such strategies in groups "2" and "3" in which the value of LC-PB exceeds 42 years, reaching values of up to 54 years. As for strategies "4" and "5", focused on increasing the building's envelope thermal resistance on the inside of the outer wall and installing thermal insulation in the existing enclosure air chamber, owing to the system's material reduction and resources applied on the construction process stage, the IRR value during the evaluation period shall be positive, reaching values of up to 15.8%. The LC-PB value of these strategies shall also be much lower, with values ranging from 10 and 27 years of return on investment. On the contrary, this type of strategy does not allow for insulating the outer facade on a continuous basis, hampering the building's thermal behaviour and reducing the NRPE use impact decrease with regards to strategies "2" and "3".

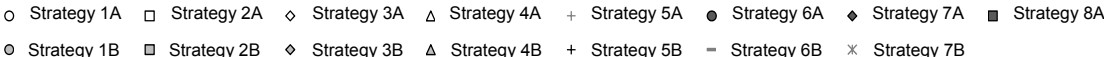

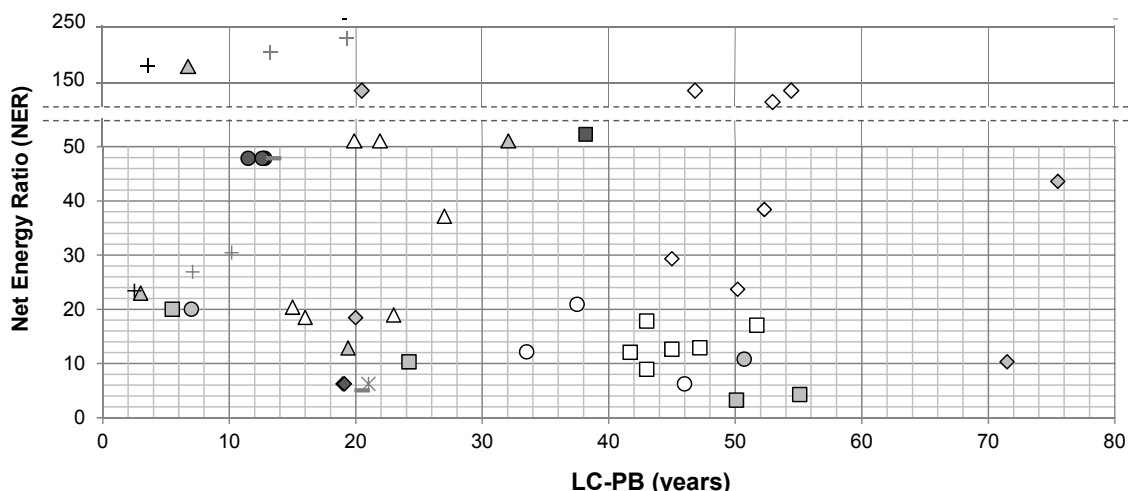

**Figure 8.** Analysis of the NER and LC-PB values of each refurbishment strategy. Numeric values for this figure are available in Table A4.

In solar thermal systems ("6"), their IRR value can reach 10% and the LC-PB value is less than 12 years. Regarding PV systems ("7"), their IRR value is 6% and their LC-PB value shall be 19 years. However, because of surface limitations when implementing these kinds of technologies (usually limited to their placement on the roofs of buildings), it is difficult to considerably reduce the building's NRPE use overall impact. For instance, all the energy generated by the two systems of this case study only reduced the baseline NRPE use impact by 11% (81 MJ/m$^2$·a). Furthermore, the results show how the values obtained by integrating a biomass boiler ("8") are suitable for reducing the NRPE use. Owing to its energy source origin, this type of power generation systems significantly reduces the amount of primary energy resource use and $CO_2$ emissions during their generation and transformation.

Finally, the analysis of the efficiency level change (evaluations based on option "B") allows to go into detail when prioritising between different options and making decisions based on their level of efficiency. The higher the thermal-energy performances of each strategy, the greater the amount of materials used, increasing the different life-cycle stages' environmental and economic impact. These results show a greater reduction in NRPE use, higher NER values, higher IRR values, and a lower LC-PB value when the efficiency level increase is positioned in basic and efficient levels. On the contrary, when positioned between efficient and advance levels, the results show a more negative scenario. These results define the difficulty to justify passive strategies with advanced levels of efficiency in moderate climates such as Donostia (Spain) and in buildings with similar characteristics. With regards to renewable systems ("6" and "7"), the growth in efficiency levels scarcely reduces their IRR value, and their NRPE use reduction percentage and LC-PB value stay in an almost straight line. Therefore, it justifies the implementation of a greater amount of this type of technology.

The NER values obtained in this study (see Figure 8) show how, while the vast majority of suggested energy refurbishment strategies have NER values higher than 10 or 20, their IRR value is negative and their LC-PB value exceeds the service life of the building subject to refurbishment. For example, in the case of strategy "3b", the NER value is 141.3, being one of the strategies with a higher NER value. However, this strategy's IRR value is negative (−0.7%) and its LC-PB value is 54.5 years. This kind of result shows the need to consider the economic dimension when prioritising refurbishment strategies and not just focusing on one of the environmental indicators.

Compared with previous studies that have analysed the NER indicator when evaluating the environmental performance of refurbishment strategies [23], the NER values obtained by this study are higher. After analysing the hypotheses considered by the study of Hernandez et al. [23], it can be seen that the reduction in the operational energy use stage was much lower, which significantly reduced the NER value.

Regarding the different refurbishment strategies, "4" and "5" shall be the highest NER options, reaching NER values of up to 231. Solar thermal systems' NER value is very positive (47.8), whereas photovoltaic panels' NER value is reduced to 6.2 owing to the high environmental impact generated mainly during the production stage.

## 5. Uncertainty of Data. Sensitivity Evaluation

The quality of the data, the rigor of each database, or the hypotheses considered directly influence the results obtained, hence limiting the potential to extrapolate the obtained results. Therefore, this methodology proposes to carry out a sensitivity analysis, where new hypotheses will be proposed in relation to some of the parameters that have the most influence when evaluating the environmental and economic behaviour of a building to be refurbished [32]. This section examines the direct influence of the uncertainty of the following parameters in the prioritisation process between different energy refurbishment strategies:

- Reference service life of the building: Because of the direct relationship between the building's assessment period and the assessed building's life-cycle impact, this study suggests two new scenarios, reducing the RSL to 25 years and increasing this value to 100 years.
- Estimated service life of material: Two additional ESL scenarios are assessed, reducing the lifetime of all products by half and increasing their life to 50 years.
- Operational energy use: Two new scenarios are added, assuming that the energy demand of the baseline scenario can be 20% higher and 20% lower than initially calculated owing to occupancy and behavioural issues.
- Conversion factor: The energy conversion factor for electricity is assumed to remain constant over a period of 50 years (RSL). If we assume renewable energy plants will be installed in these coming decades, the primary energy conversion factor for electricity is very likely to decrease. However, energy markets and policies rely on various global geopolitical factors, and this issue is very difficult to predict. Therefore, two new scenarios are added to take these issues into account, assuming that the conversion factors of the baseline scenario can be 20% higher and 20% lower than initially calculated.
- Energy price (EPI): Two new scenarios are added, assuming that the energy prince increment defined in the baseline scenario can be zero percent (0%) or double that initially calculated.
- Inflation rate: Two additional scenarios are added to take into account uncertainties in the complex economic markets, assuming that the inflation value can be 0% and 3%.

### 5.1. Results of the Sensitivity Analysis

This section shows the percentage influence of each of these parameters on the previous results. The 0% axis shall draw the line of the results obtained by means of the original inputs defined for

the case study and, from that point, the influence of uncertainty on the proposed parameters shall be evaluated.

Figure 9 shows how the NRPER value could vary ±30%, reaching a maximum variation of −44%. This variation is mainly linked to the direct influence of three parameters. The first critical parameter is the operational energy demand uncertainty, whose influence can alter the results by up to ±25%. Finally, the second parameter is connected with the conversion factors applicable when calculating the environmental impact on the different energy sources. According to the source of the data or the adaptation of each factor to each member state, results can vary up to ±25%. In the case of Spain, there is an ever-growing number of studies attempting to upgrade and integrate the LCA concept in such a calculation [80]. However, there still are substantial uncertainties when it comes to making a proper calculation with regards to this factor, with such an influence on the final results. Figure 9 shows how the NER value might vary by ±50%, reaching a maximum variation of 100%. Values of the RSL of the building and the ESL of each refurbishment strategy will be directly related to the NER results. Moreover, the lower the RSL value, the lower the potential reduction generated during the building's use stage, leading to a reduction of the different strategies' NER value by 50%. Finally, although the influence of the parameters related to the operational energy demand and conversion factor uncertainty is lower, it is difficult not to take them into account, as an inadequate definition of these two parameters may cause the NER value to vary by ±20%.

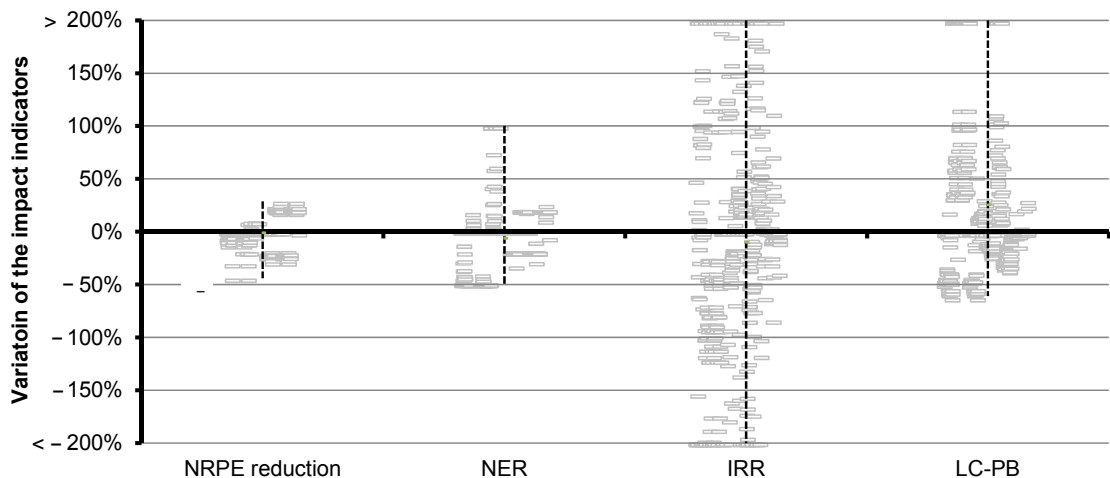

**Figure 9.** Variation of the NRPE use reduction, NER, IRR, and LC-PB values owing to the new scenarios defined by the sensitivity assessment.

As shown in the results of Figure 9, the IRR economic indicator variation result surpasses the ±200% barrier, reaching values of up to 450% or −320%. That is, it is difficult to accurately determine the profitability of each refurbishment strategy owing to uncertainty of aspects such as the ESL of the products used, the RSL of the refurbished building, the building's energy demand, or the energy's rising price during the refurbished building's life cycle. Regarding the inflation factor, its influence is not as relevant in those strategies with low maintenance costs in comparison with other parameters (except in renewable systems). Finally, although the variation in results is less than that of the IRR economic indicator, Figure 9 shows that the LC-PB value may vary by ±50% depending on the input or uncertainty data quality. This value can be significantly increased, reaching values of up to 300% in scenarios where the EPI is zero and the ESL value of the refurbishment strategy is reduced. On the other hand, in scenarios suggesting significant increases of the energy price or in which the strategies' ESL value is increased, the value of the investment return period may be reduced by 60%.

### 5.2. Example of One of the Refurbishment Strategies

In order to show the relevance and direct influence of the uncertainty in determining the inputs of some parameters, this last section develops, in detail, a life-cycle cost assessment of the case study after being refurbished by strategy "2b" (application of a new ventilated façade with basic efficiency level). Together with the results obtained from original inputs (defined during the development of this work), this exercise offers three new input data for two parameters that directly influence the calculation methodology:

- Double energy price increment: 8% a year for electricity and 6% a year for natural gas.
- Energy price increment: 0%.
- Reduction of the estimated service life value (ESL) of each product that composes this strategy in half.

Figure 10 shows how the inputs totally change the interpretation of the results of the life-cycle cost evaluation of the refurbished building.

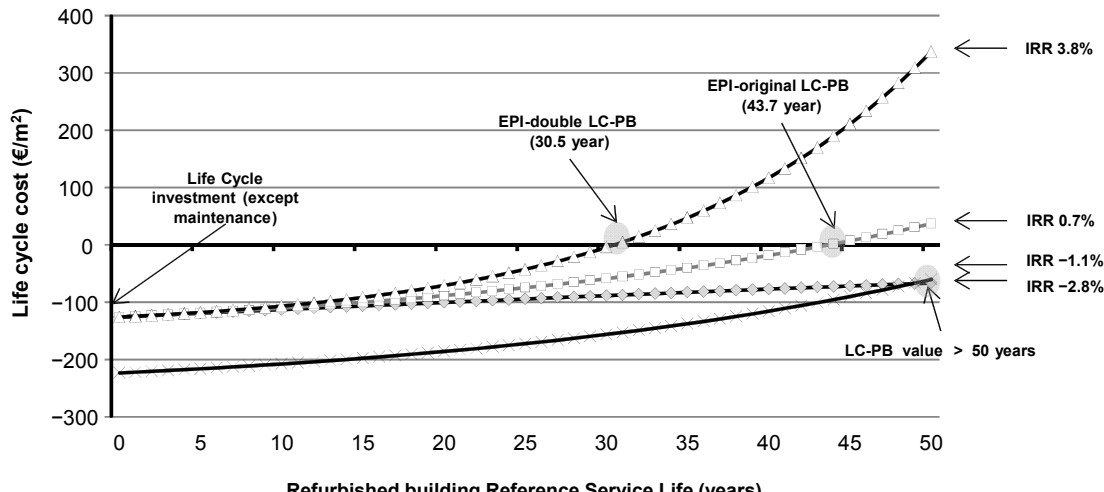

**Figure 10.** Life-cycle cost assessment of the building owing to the application of the "2bl" refurbishment strategy. EPI, energy price.

At the initial point of refurbishment (year 0), based on the criteria proposed for this calculation methodology, the life-cycle investment will be the same in three of the scenarios ("0/a/b"). However, if the user proposes to reduce the ESL value of the products, the initial point (life-cycle investment) is greater in option "c" because of the increased number of replacements during the life of the refurbished building. From this initial point, according to the different proposals of the increase in energy price, the reduction of the economic impact during the operational stage varies, generating three different trends.

The first trend defined by option "a" proposes a scenario where the price increase of the energy is very high. In this case, the IRR value of the investment will be 3.8% with an LC-PB of 30.5 years. However, owing to different reasons unrelated to this study, in the case that the price increase of the energy will be null ("b"), the profitability of the refurbishment strategy will be negative (−2.8%) and the value of LC-PB will exceed the value of the refurbished building's RSL. Between these two tendencies is located the original scenario, whose profitability is 0.7% and the LC-PB value is nearly 44 years. Regarding "c" option, owing to its high impact related to the replacement stage (B4), the IRR value decreases (−1.1%) and the return on investment exceeds the barrier of life of the refurbished building (>50 years).

## 6. Conclusions

The results from the analysis of the refurbishment strategies for the case study validate the need to apply this type of indicator in a unified way, because the use of one or two indicators in isolation would obviate much of the information that it is necessary to consider when prioritizing between different refurbishment strategies.

The results show that, in those refurbishment strategies with a higher reduction rate of NRPE use, their IRR value is very low (even negative in many cases) and their LC-PB value is very high. In fact, in some cases, it is even higher than the RSL value of the building to be refurbished. The NER values obtained in this case study show how, whereas the vast majority of suggested energy refurbishment strategies have NER values higher than 10 or 20, their IRR value is negative and their LC-PB value exceeds the service life of the building subject to refurbishment.

With regards to the analysis of the efficiency level of the strategies, the results show a greater reduction in NRPE use and IRR values and a lower LC-PB value when the efficiency level increase is positioned in basic and efficient levels. On the contrary, when positioned between efficient and advanced levels, the results show a more negative scenario. Furthermore, NER values for efficiencies between the basic and efficient levels are much higher than those obtained from the efficient-advanced level, thus making it difficult to justify strategies with advanced levels of efficiency in moderate climates such as Donostia (Spain) and in buildings with similar characteristics.

Regarding the sensitivity assessment, this study reflects the high uncertainty on some of the parameters and their high influence on the final results. After analysing new scenarios related to parameters such as reference service life of the building, estimated service life of material, operational energy use, conversion factor, energy price, and inflation rate, the results shows how the NRPE use reduction value could vary up to −44%. The variation of the other indicators is also very relevant, reaching variation rates such as 100% for the NER, 450% for the IRR, and 300% for the LC-PB. Together with the analysis of the variation of the results, this study has allowed to determine which are the parameters or inputs that most influence each indicator, which is relevant information when calibrating the prioritization process for the refurbishment strategy.

**Author Contributions:** Conceptualization, X.O.; methodology, X.O., R.J.H., and P.H.; validation, X.O.; investigation, X.O. and P.H.; resources, X.O.; data curation, X.O. and R.J.H.; writing—original draft preparation, X.O.; writing—review and editing, X.O., R.J.H., and P.H.; supervision, R.J.H. and P.H. All authors have read and agreed to the published version of the manuscript.

**Funding:** This research received no external funding.

**Acknowledgments:** The authors thank the Department of Architecture and the Vicerrectorate for Research of the University of the Basque Country UPV/EHU for the financial support given for this research.

**Conflicts of Interest:** The authors declare no conflict of interest.

## Appendix A

This appendix shows all the information related to the refurbishment strategies considered by this study. Eight different refurbishment strategies were considered and evaluated:

1. Replacement of all existing windows with a new frame and glazing.
2. External thermal improvement by a ventilated facade system. Three layers composed this system: the aluminium substructure, the insulation, and the ceramic out-layer.
3. External thermal improvement by an external insulation system. Two layers composed this system: the insulation and the mortar out-layer.
4. Indoor thermal improvement solution. Two layers composed this system: the insulation and the plasterboard.
5. Air chamber insulation.
6. Solar thermal system on the roof of the building, which uses solar energy to generate heat that is then used to produce hot water for dwelling hot water (DHW).

7.　　Photovoltaic panels on the roof of the building, generating and exporting electricity from a renewable source to the national grid.

8.　　Biomass boiler for the heating system.

Within the refurbishment strategies (except for the eighth strategy), three different levels of efficiency are proposed: basic, efficient, and advanced (see Table A1). The basic efficiency level is based on minimum thermal criteria determined by existing Spanish regulations. The efficient efficiency level is based on improving the thermal properties and the renewable generation by 30% compared with the basic level. Finally, the advanced level improves the thermal properties of the building to very high values, such as those used in standards as the passive house [81]. In addition, this level considers increasing renewable energy generation by 50% compared with the basic level.

The combination between the number of each strategy and the efficiency level makes possible the definition of the acronym of each refurbishment strategy. For example, a ventilated façade with efficient level will be a "2e".

**Table A1.** Set of energy refurbishment strategies applied in this case study.

| Efficiency Level | Refurbishment Strategies | | | | | | | |
| --- | --- | --- | --- | --- | --- | --- | --- | --- |
| | 1 | 2 | 3 | 4 | 5 | | 6 | 7 |
| | Composition and Thermal Property of the Window | Insulation Layer Thickness (Centimetres) | | | | | Panel Area (m$^2$) | |
| Basic (b) | Double glazed + aluminium frame (2.7 W/(m$^2$K)) | Facade 5 cm; deck 8 cm; first floor slab 6 cm | | | | | 75 | Not mandatory |
| Efficient (e) | Double glazed low-e + PVC frame (2.0 W/(m$^2$K)) | Facade 9 cm; deck 13 cm; first floor slab 10 cm | | | | | 110 | 120 |
| Advance (a) | Triple glazed + wood frame (1.4 W/(m$^2$K)) | Facade 25 cm; deck 30 cm; first floor slab 15 cm | | | | | 140 | 400 |

Along with these 20 refurbishment strategies, which will make it possible to evaluate the impact reduction of the refurbished building compared with the baseline ("A" option according to the Figure A1), this study proposes to generate new refurbishment scenarios to analyse the influence of the increase in the level of efficiency between each strategy ("B" option according to the Figure A1). For this purpose, there are 12 new scenarios defined and their NRPER, NER, IRR, and LC-PB indicators values are calculated.

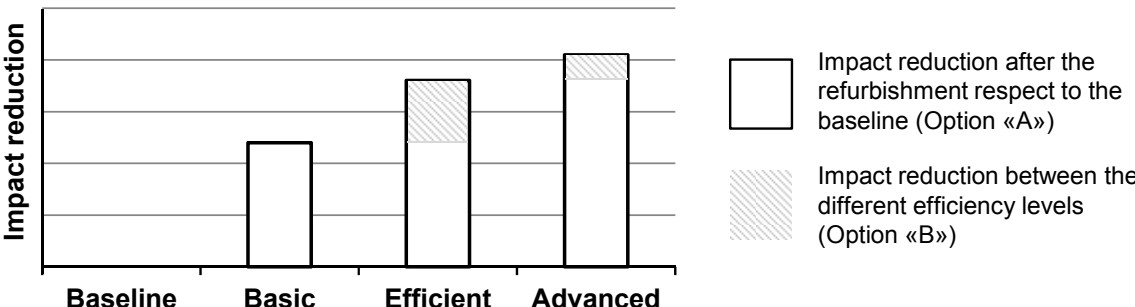

**Figure A1.** Scheme of the refurbishment efficiency level optimization.

Following, Table A2 shows the non-renewable primary energy use, transported distance (Dm), and estimated service life (ESL) of the products and systems applied by the different refurbishment strategies.

Finally, Table A3 defines the economic data related to each refurbishment strategy.

**Table A2.** Environmental data and hypothesis considered in relation to the product and systems applied by the refurbishment strategies.

| Name | Source | Unit | NRPE (MJ/Unit) | Dm (km) | ESL (Years) |
|---|---|---|---|---|---|
| Double glazed | [82] | $m^2$ | $4.63 \times 10^2$ | 120 | 30 [82] |
| Double glazed low-e | [82] | $m^2$ | $6.21 \times 10^2$ | 120 | 30 [82] |
| Triple glazed | [82] | $m^2$ | $7.89 \times 10^2$ | 300 | 30 [82] |
| Aluminium frame | Window frame, aluminium, at plant [71] | $m^2$ | $7.22 \times 10^3$ | 120 | 20 [83] |
| PVC frame | Window frame, plastic (PVC), at plant [71] | $m^2$ | $5.63 \times 10^3$ | 120 | 30 [83] |
| Wood frame | Window frame, wood, U = 1.5 W/m$^2$·K, at plant [71] | $m^2$ | $2.32 \times 10^3$ | 300 | 30 [83] |
| Aluminium sub-structure | CERTIFIED. Extruded aluminium industry-average | kg | $7.13 \times 10^1$ | 50 | 50 [83] |
| Out-layer | Ceramic façade panels. Ceramic PE (2008) [69] | $m^2$ | $2.56 \times 10^2$ | 120 | 50 [83] |
| Mortar | Cement mortar, at plant [71] | kg | $1.49 \times 10^0$ | 50 | 35 [83] |
| Plasterboard | Gypsum plaster board, at plant [71] | kg | $5.75 \times 10^0$ | 120 | 30 [83] |
| Solar thermal panel | Flat plate collector, at plant [71] | $m^2$ | $1.50 \times 10^3$ | 300 | 30 [84] |
| Photovoltaic panel | Photovoltaic panel, mono-Si, at plant [71] | $m^2$ | $3.33 \times 10^3$ | 300 | 30 [84] |
| Boiler | Biomass boiler, Europe (RER) [71] | unit | $6.32 \times 10^3$ | 120 | 30 [83] |

**Table A3.** Economic data of the refurbishment strategies.

| Refurbishment Strategy ID | Source | Unit | A-13 Stage (€/Unit) | A5 Stage (€/Unit) | B2 Stage (€/Unit·A) |
|---|---|---|---|---|---|
| 1b | Verband der Fenster und Fassadenhersteller [85] | | 198 | 104 | 0.3 |
| 1e | | | 217 | 104 | 0.3 |
| 1a | | | 380 | 104 | 0.3 |
| 2b | ECOFYS [86] and Generador de precios [87] | $m^2$ | 154 | 47 | 0.4 |
| 2e | | | 157 | 47 | 0.4 |
| 2a | | | 202 | 51 | 0.4 |
| 3b | | | 83 | 26 | 0.6 |
| 3e | | | 92 | 26 | 0.6 |
| 3a | | | 143 | 30 | 0.6 |
| 4b | | | 38 | 11 | 0.1 |
| 4e | | | 42 | 11 | 0.1 |
| 4a | | | 61 | 11 | 0.1 |
| 5b | | | 30 | 15 | – |
| 5e | | | 33 | 15 | – |
| 6 | Solar District Heating Guidelines [88] | | 437 | 113 | 29.5 |
| 7 | Smestad [89] | | 265 | 14 | 6.9 |
| 8 | [87] | | 35,000 | 1350 | 68.5 |

## Appendix B

Numeric value of the non-renewable primary energy use reduction (NRPER), net energy ratio (NER), internal rate of return (IRR), and life-cycle payback (LC-PB) for each refurbishment scenario defined by Appendix A.

**Table A4.** Impact for each refurbishment scenario.

| Option | Refurbishment Strategy ID | NRPER (MJ/(m²·a)) | NER | IRR (%) | LC-PB (Years) |
|---|---|---|---|---|---|
| A | 1A-b | 47.4 | 6.2 | 0.4 | 46.0 |
| | 1A-e | 93.2 | 12.1 | 2.3 | 33.5 |
| | 1A-a | 120.5 | 20.9 | 1.6 | 37.5 |
| | 2A-b | 86.8 | 17.1 | −0.3 | 51.7 |
| | 2A-e | 122.0 | 17.8 | 0.8 | 43.0 |
| | 2A-a | 136.7 | 12.6 | 0.5 | 45.0 |
| | 3A-b | 91.5 | 141.3 | −0.7 | 54.5 |
| | 3A-e | 128.3 | 141.0 | 0.3 | 47.0 |
| | 3A-a | 147.1 | 109.4 | −0.5 | 53.0 |
| | 4A-b | 59.0 | 37.1 | 3.7 | 27.0 |
| | 4A-e | 100.3 | 54.6 | 5.7 | 20.0 |
| | 4A-a | 121.4 | 54.6 | 5.0 | 22.0 |
| | 5A-b | 54.9 | 231.8 | 5.9 | 19.2 |
| | 5A-e | 97.0 | 205.0 | 8.8 | 13.4 |
| | 6A-b | 26.0 | 47.8 | 10.8 | 11.5 |
| | 6A-e | 38.1 | 47.8 | 10.0 | 12.8 |
| | 6A-a | 48.5 | 47.8 | 10.0 | 12.6 |
| | 7A-e | 9.8 | 6.2 | 6.6 | 19.0 |
| | 7A-a | 32.8 | 6.2 | 6.6 | 19.1 |
| | 8A | 296.2 | 69.0 | 1.5 | 38.0 |
| B | 1B-b/e (basic to efficient) | 45.8 | 20.0 | 16.2 | 7.0 |
| | 1B-e/a (efficient to advanced) | 27.2 | 10.8 | −0.2 | 50.7 |
| | 2B-b/e | 35.2 | 20.0 | 20.2 | 5.5 |
| | 2B-e/a | 14.7 | 4.3 | −0.7 | 55.1 |
| | 3B-b/e | 36.8 | 140.3 | 5.6 | 20.2 |
| | 3B-e/a | 18.8 | 43.6 | −2.8 | 75.5 |
| | 4B-b/e | 41.3 | 174.8 | 17.9 | 6.2 |
| | 4B-e/a | 21.1 | 54.3 | 3.0 | 30.2 |
| | 5B-b/e | 42.1 | 178.1 | 30.2 | 3.5 |
| | 6B-b/e | 12.1 | 47.9 | 6.2 | 20.5 |
| | 6B-e/a | 10.4 | 47.8 | 8.7 | 13.5 |
| | 7B-e/a | 22.9 | 6.2 | 5.4 | 21.0 |

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
