# Peer review of "Environmental and Economic Prioritization of Building Energy Refurbishment Strategies with Life-Cycle Approach"

_sustainability, doi:10.3390/su12093914_

Round 1

Reviewer 1 Report

This work is very interesting and innovative.

As underlined in the paper, one of the most crtical aspects is the high variability of values, so it would be important to continue to study this topic.

Author Response

Dear Editor,
We would like to thank you and the reviewers again for your comments on the submission of our
manuscript \Two-stage Multi-objective Meta-heuristics for Environmental and Cost-Optimal Energy Re-
furbishment at District Level". We truly think that this new version of our paper addresses carefully all
the reviewer's concerns. Please nd below our detailed response to such comments.
Thank you very much for your time and attention.
Yours sincerely,
The authors.

REVIEWER 1.

The authors appreciate the comment of the reviewer

Reviewer 2 Report

Too many single "strategies". Combinations of the single refurbishments can decrease the number of results.

Author Response

Response to Reviewer 2 Comments

Too many single "strategies". Combinations of the single refurbishments can decrease the number of results

The authors are aware that the number of strategies is very high. However, in order to analyze the environmental or economic behavior of each one of them, and in turn, in order to analyze how the variation of some parameters influences the final results (sensitivity analysis), the authors propose to maintain the number of current strategies.

Reviewer 3 Report

The paper presents four indicators that can help stakeholders to determine the rehabilitation strategies to improve the environmental performance and ensure economic returns. These indicators are :

  1. Non-Renewable primary energy use reduction 

  2. Net Energy Ratio

  3. Internal return rate 

  4. Life cycle payback

The study is interesting as it is aimed to help stakeholders narrow down the rehabilitation choices with respect to minimizing environmental impacts and reducing economic costs. However, the study can be considerably improved if the manuscript is better structured. Currently the key methods and results the study points to are complicated to infer and with some changes this could be significantly improved.

Here are my main comments:

1) The introduction can be improved with more description on the use of the mentioned indicators, including if and how previous studies have used them. It should finally end with a clear goal and scope of the study, which is currently missing. 

2) Table 1 refers to two columns as environmental indicators. I reckon one of them is economic. 

3) Ln 71 Avoid referring to the indicators as 'impact indicators'. Instead use 'indicators'

4) The methodology (section 2.1)- Simplify the description. First introduce the stages as given in the standard EN15978:2011. How are they typically applied to new construction/baseline. Then how are these stages applied to rehablitation strategies.

5) Figure 2 - explain each scenario in detail. Although with respect to the paper I suggest add the figure and information in the supplementary material with respect to choice of method for calculating payback .

6) Section 2.3- Interpretation of results - This section should be placed before the results section. Moreover the text is currently too convoluted. recommend referring only to the unified method. why this was chosen over other result visualization method can be discussed in the discussion section.

7) Section 3- More description could be added with respect to the rehabilitation strategies applied to this case study.

8) Table 2 -  Why were two functional units selected? The results of this study are based on which functional unit.

9) Table 2- What are the estimated service life - provide detail.

10) Ln 199 and Ln 203 seem like the two goals of this study. Must have been mentioned at the end of the introduction.

11) Fig 6 - 7 preferably avoid using multiple Y axis (especially when the X axis does not contain categorical data. Its harder to read and interpret.)

12) What are the parameterized strategies? Recommend expanding on the strategies and their parameters in the section with case study.

13) Section 5 on sensitivity analysis - Add this to methodology / case study section.

14) Line 295-297 Incorrect method of performing hybrid Life cycle assessment. Hybrid life cycle assessment is performed by combining the process based study to an Input Output table (see example Ewing et al. https://doi.org/10.1111/j.1530-9290.2011.00374.x )

15) Line 343 'some of the parameters' - be more specific

Minor comments:

1) Prefer using the term 'refurbishment' or 'renovation' rather than 'rehabilitation'

2)  Line 13 - Remove 'also'

4) Line 47 - Remove ' Among others, the authors'

5) Section 2 is referred to as 'Calculation methodology and hypothesis'. There is no hypothesis presented. Recommend re-phrasing to 'Methodology'.

6) Delete line 212 'Once all calculations.... have been made'

7) Proof reading recommended

Author Response

Response to Reviewer 3 Comments

1) The introduction can be improved with more description on the use of the mentioned indicators, including if and how previous studies have used them. It should finally end with a clear goal and scope of the study, which is currently missing. 

The authors have improved the introduction, integrating the indicators that are going to be used by this study and also, defining more clearly the goal and scope of the study.

Related to the "more description on the use of the mentioned indicators" suggestion, after doing a first test where the possibility of integrating the detailed description of each indicator in this first section has been considered, the authors have ruled out this possibility and they proposed to keep the detailed description of each indicator in section 2.2. They are aware that this information could be useful in the introduction. However, seeing the length of the description of some indicators (mainly the LC-PB), the authors have decided to keep all this information in section 2.2, since otherwise the introduction be too extensive.

Related to the "description on the indicators, including if and how previous studies have used them" suggestion, the authors have improved the section 2.2 with this information

2) Table 1 refers to two columns as environmental indicators. I reckon one of them is economic. 

It has been corrected

3) Ln 71 Avoid referring to the indicators as 'impact indicators'. Instead use 'indicators'

The concept “impact indicators” has been replaced by “indicators” throughout the document

4) The methodology (section 2.1)- Simplify the description. First introduce the stages as given in the standard EN15978:2011. How are they typically applied to new construction/baseline. Then how are these stages applied to rehablitation strategies.

On the one hand, the authors have cited the life-cycle stages defined by the standard EN 15978:211.

Furthermore, the authors have added a new paragraph to describe which life-cycle stages have been applied by previous studies and which life-cycle stages are the most relevant in a refurbishment project.

Finally, the authors have tried to simplify the description of the methodology.

5) Figure 2 - explain each scenario in detail. Although with respect to the paper I suggest add the figure and information in the supplementary material with respect to choice of method for calculating payback .

On the one hand, the authors have added a new paragraph to describe each scenario

On the other hand, after analyzing the possibility of adding the figure and information in the supplementary material, the authors consider that this critical review of the "payback" indicator is relevant to justify the added value of the Life-cycle payback indicator. Therefore, the authors suggest maintaining this description and the figure as part of the main article.

6) Section 2.3- Interpretation of results - This section should be placed before the results section. Moreover the text is currently too convoluted. recommend referring only to the unified method. why this was chosen over other result visualization method can be discussed in the discussion section.

Regarding the placement of this section, the authors consider that this section is correctly placed (the results are presented during the section 4).

The authors have considered the suggestion of the reviewer and they have defined only the unified method.

Finally, the selection of the unified method has been justified by the result-discussion section.

7) Section 3- More description could be added with respect to the rehabilitation strategies applied to this case study.

The authors have added all the information related to the refurbishment strategies considered by the study: description, efficiency levels, environmental data, economic data, processes... Due to the high amount of information related to this issue, the authors have decided to integrate all this information into a new Appendix, out of the main article.

8) Table 2 -  Why were two functional units selected? The results of this study are based on which functional unit.

The functional unit is unique (annual impact per each living floor square meter). For example, if the impact is calculated by the indicator “primary energy”, the results are going to be showed in “MJ/(m2·a)”. That is, this functional unit allow showing annualized results and also, considering the living floor area of the building.

Based on this clarification, the authors consider that the current Functional unit is correct

9) Table 2- What are the estimated service life - provide detail.

In order to show the estimated service life of each refurbishment strategy, the authors have added this data by the “Appendix A”.

10) Ln 199 and Ln 203 seem like the two goals of this study. Must have been mentioned at the end of the introduction.

By this section the authors tried to explain how they considered some new scenarios to assess the relevance of the efficiency level of each strategy. However, as the reviewer suggest, this section creates confusion and it seems like two different goals. Therefore, the authors suggest to integrate this information into the new Appendix A, where it is going to be described all the information related to the refurbishment strategies.

11) Fig 6 - 7 preferably avoid using multiple Y axis (especially when the X axis does not contain categorical data. Its harder to read and interpret.)

The authors have deleted the second Y axis in Figures 6 and 7.

12) What are the parameterized strategies? Recommend expanding on the strategies and their parameters in the section with case study.

The use of the word "parameterized" had been incorrectly used. Therefore, the authors have deleted this concept. Additionally, in order to explain in detail the refurbishment strategies that are going to be assessed, it is added a new Appendix A

13) Section 5 on sensitivity analysis - Add this to methodology / case study section.

The authors have added a new sub-section into the methodology section describing the need of the sensitivity assessment. However, the authors feel that the new hypotheses considered and the results obtained after this sensitivity assessment are adequately integrated into the section 5.

14) Line 295-297 Incorrect method of performing hybrid Life cycle assessment. Hybrid life cycle assessment is performed by combining the process based study to an Input Output table (see example Ewing et al. https://doi.org/10.1111/j.1530-9290.2011.00374.x )

The authors have improved this section, correcting the error defined by the reviewer.

15) Line 343 'some of the parameters' - be more specific

During the second part of this paragraph the authors have described exactly what are the parameters that most influence the economic sensitivity analysis. Therefore, according to the reviewer's observation, it is proposed to delete the first sentences of this paragraph

Minor comments:

1) Prefer using the term 'refurbishment' or 'renovation' rather than 'rehabilitation'

It is solved

2)  Line 13 - Remove 'also

It has been removed

4) Line 47 - Remove ' Among others, the authors'

It has been removed

5) Section 2 is referred to as 'Calculation methodology and hypothesis'. There is no hypothesis presented. Recommend re-phrasing to 'Methodology'.

It has been changed

6) Delete line 212 'Once all calculations.... have been made'

It has been deleted

7) Proof reading recommended

The authors have tried to improve the structure, content and grammar of the paper

Reviewer 4 Report

The paper evaluates the indicators for analyzing the rehabilitation project of residential building with life-cycle approach. The topic of this article is interesting, but the paper needs improvements to match the journal standard. The detailed comments are follows:

In abstract, there are many rudimental information. Please consider only stating things once and stating them well.

The definition and meaning of indictors and parameters used should be given and clarified. Hypothesis and assumptions of this study should be also presented.

The importance and reason of cases selected should be added.

The obtained result should be compared with previous works for validation.

Author Response

Response to Reviewer 4 Comments

In abstract, there are many rudimental information. Please consider only stating things once and stating them well.

Based on the suggestion of the reviewer, the authors have simplified the abstract, considering only stating things once.

The definition and meaning of indictors and parameters used should be given and clarified.

One the one hand, the authors have included the indicators that are going to be considered by this study into the introduction section. Additionally, the authors have improved the section 2.2, adding more information about the indicators considered by this study.

Hypothesis and assumptions of this study should be also presented.

The authors have completed the Table 2, defining most of the hypothesis and assumptions related to the assessment process. Furthermore, it is import to highlight that in order to simplify this table; the authors have proposed to add a new Appendix A with all the information related to each refurbishment strategy.

The importance and reason of cases selected should be added.

Based on the suggestion of the reviewer, the authors have added an introduction to justify the replicability potential of the typology selected by the study

The obtained result should be compared with previous works for validation.

The authors have compared the results with some provious studies

Round 2

Reviewer 3 Report

Pg 15 ln 445 Hybrid LCI cannot be implemented by multiplying the results by a co-efficient . This is simply incorrect. Hybrid LCI is implemented when one combines the process based data to an Input-Output dataset. The background data is different. I recommend the authors to remove this part of the analysis if they don't know the methodology.

Functional unit- Please consider writing 'sq metre Useful Floor Area'

Author Response

Response to Reviewer 3 Comments

Pg 15 ln 445 Hybrid LCI cannot be implemented by multiplying the results by a co-efficient . This is simply incorrect. Hybrid LCI is implemented when one combines the process based data to an Input-Output dataset. The background data is different. I recommend the authors to remove this part of the analysis if they don't know the methodology.

The authors appreciate the clarification of the reviewer. Therefore, based on this suggestion, the authors have removed this part of the analysis and have adapted the "5.1. Results of the Sensitivity Analysis" section (content and figure 9)

Functional unit- Please consider writing 'sq metre Useful Floor Area'

The authors have considered writing the functional unit suggested by the reviewer (see Table 2)

Reviewer 4 Report

The reviewer has examined the revision made by the authors. It is shown that most of them are addressed and the comments are reflected in the revised manuscript. The followings are given for further revision.

In the Section 2 of methodology, the interpretation results and sensitivity assessment should not be included. Only the technique, procedure and analysis method will be presented.

What is the difference between Case study and Results and discussion? Please, clarify the section name properly and scholarly.

Author Response

Response to Reviewer 4 Comments

In the Section 2 of methodology, the interpretation results and sensitivity assessment should not be included. Only the technique, procedure and analysis method will be presented.

The authors agree with the reviewer that the interpretation of the results should not be within the methodology. Therefore, they have completed this information within section 4 (results and discussion).

Regarding the sensitivity analysis, this section was added within the methodology (section 2) during the first review based on the suggestion of the reviewer 3: “Comment 13: Section 5 on sensitivity analysis - Add this to methodology / case study section”. However, the authors agree that the description of the sensitivity assessment should not form part of this section. Therefore, this information has been removed. 

What is the difference between Case study and Results and discussion? Please, clarify the section name properly and scholarly.

The authors also had the question whether it was necessary to integrate these two sections into a single section or whether it was better to separate it into two (as they do now). After analyzing other articles published in the journal or in other high-impact journals, they decided to separate all that information into two.

On the one hand, the "case study" section focuses on describing the case study and all the hypotheses related to the calculation process. On the other hand, the "results and discussion" section shows and evaluates the obtained results.

After analyzing the reviewer's suggestion, the authors propose to:

  • Complete the name of section 3: "case study and hypothesis".
  • Maintain the name of section 4: "results and discussion"

Round 3

Reviewer 4 Report

The reviewer's comments have been reflected in the revision.

Author Response

Response to Reviewer 4 Comments

The reviewer's comments have been reflected in the revision.

The authors appreciate the comment of the reviewer